# Plant Diversity and Agroecosystem Function in Riparian Agroforests: Providing Ecosystem Services and Land-Use Transition

**Serra W. Buchanan [1], Megan Baskerville [2], Maren Oelbermann [2], Andrew M. Gordon [3], Naresh V. Thevathasan [3] and Marney E. Isaac [1,\*]**

[1] Department of Physical and Environmental Science, University of Toronto Scarborough, Toronto, ON M1C 1A4, Canada; serrawillow.buchanan@mail.utoronto.ca

[2] Department of Environment, University of Waterloo, Waterloo, ON N2L 3G1, Canada; megan.baskerville@canada.ca (M.B.); moelbermann@uwaterloo.ca (M.O.)

[3] School of Environmental Sciences, University of Guelph, Guelph, ON N1G 2W1, Canada; agordon@uoguelph.ca (A.M.G.); nthevath@uoguelph.ca (N.V.T.)

\* Correspondence: marney.isaac@utoronto.ca

**Abstract:** Achieving biologically diverse agricultural systems requires a commitment to changes in land use. While in-field agrobiodiversity is a critical route to such a transition, riparian systems remain an important, yet understudied, pathway to achieve key diversity and ecosystem services and targets. Notably, at the interface of agricultural landscapes and aquatic systems, the diversification of riparian buffers with trees reduces the non-point source pollution in waterways. However, in riparian agroforestry systems, little is known about herbaceous community patterns and, importantly, the herbaceous community's role in governing carbon (C) and nitrogen (N) cycling. Our study investigated herbaceous community taxonomic and phylogenetic diversity patterns in riparian (i) grasslands (GRASSLAND), (ii) rehabilitated agroforests (AGROFOREST-REHAB), and (iii) remnant forests (AGROFOREST-NATURAL). We then determined the biodiversity-ecosystem function relationships between community functional diversity metrics, C and N cycling, and greenhouse gas fluxes. We observed significant differences in taxonomic and phylogenetic diversity among riparian buffer types. We found that herbaceous plant communities in riparian agroforestry systems expressed plant trait syndromes associated with fast-growing, resource acquiring strategies, while grassland buffer plants exhibited slow-growing, resource conserving strategies. Herbaceous communities with high functional diversity and resource acquiring trait syndromes, such as those in the agroforestry riparian systems, were significantly correlated with lower rates of soil $CO_2$ efflux and N mineralization, both of which are key fluxes related to ecosystem service delivery. Our findings provide further evidence that functionally diverse, and not necessarily taxonomically diverse, plant communities are strongly correlated to positive ecosystem processes in riparian agroforestry systems, and that these communities contribute to the transition of agricultural lands toward biologically and functionally diverse landscapes.

**Keywords:** agrobiodiversity; functional diversity; greenhouse gas fluxes; plant functional traits; riparian agroforestry; soil $CO_2$ efflux; soil N mineralization

## 1. Introduction

Increasing the biological complexity of agroecosystems can substantially enhance ecosystem services at regional and global scales [1,2]. While these diversified agroecosystems can take a number of different forms (e.g., crop rotation and intercropping), riparian systems remain a rather unexplored and understudied element of diversified agroecosystems, even though these systems have proven to be

a popular and well received form of land diversification [3,4]. While it has been well established that vegetated riparian buffers can reduce non-point source inputs (sediment, nutrients, and pesticides) into waterways [5–7], these buffers have the potential to perform a number of other ecosystem functions. Plant communities within these systems have an influence on carbon (C) and nitrogen (N) cycling processes either directly through root uptake and plant chemical cycling, or indirectly through the supply of litter and the stabilization of nutrients in the soil [8,9]. Depending on biotic and abiotic conditions, riparian buffers have also been shown to act as a source or sink for greenhouse gases [10–12], such as carbon dioxide ($CO_2$), nitrous oxide ($N_2O$), and methane ($CH_4$), and play an important role in nutrient transformation in the form of N mineralization [8]. Arguably, riparian agroforestry buffers are essential local and regional zones of watershed purification, biodiversity hotspots, and carbon sinks. However, the delivery of these services is presumably highly related to the specific plant communities.

Plant community biodiversity and ecosystem functioning relationships have relied on a range of diversity indices, including taxonomic, phylogenetic, and functional [13–15]. In particular, aboveground leaf traits have been used to describe resource use strategies by plants, where leaf traits fall along a spectrum from resource acquisition to conservation, known as the Leaf Economics Spectrum (LES) [16]. Individual plant-level traits can then be extrapolated to provide insight on C and N cycling processes at the community and ecosystem level [17]. The extent to which plant traits govern ecosystem processes rely on two hypotheses: the mass-ratio hypothesis [18] states that ecosystem processes are mediated by the most dominant functional traits of a plant community [19], and niche partitioning states that diversity within these traits will also underpin plant functioning as a result of complementarity [14,20]. Management decisions about riparian buffer rehabilitation and regeneration will directly alter herbaceous plant communities and the traits expressed within these communities, which in turn will likely influence C and N cycling processes [6].

In this study, we examine the diversity of plant communities in riparian agroforestry systems and the consequences for localized soil C and N dynamics. Specifically, using species composition and leaf traits, we determine the taxonomic, phylogenetic, and functional diversity indices of herbaceous plant communities in three common riparian buffers types in a Southern Ontario, Canada, watershed. We pair this with in-situ measurements of soil gas emissions ($CO_2$, $N_2O$, and $CH_4$) and rates of soil N mineralization. We aim to (1) assess taxonomic, phylogenetic, and functional diversity within common riparian buffer types; (2) assess the functional trait syndromes of herbaceous communities; and (3) measure key soil processes related to ecosystem function and services.

## 2. Materials and Methods

### 2.1. Site Description and Study Design

Our study was conducted within agricultural riparian buffers along Washington Creek (43°18′ N, 80°34′ W), a spring-fed first order stream within the Grand River watershed in Southern Ontario, Canada. In the summer of 2017 and 2018, we sampled three riparian buffer types: pastureland, which had previously (over 30 years ago) been fenced off and returned to a natural grassland (GRASSLAND), a rehabilitated site, which was planted in 1985 at a density of 3.14 trees m$^{-2}$ with poplar hybrids (*Populus spp.*) as well as silver maple (*Acer saccharinum*) and Carolina poplar (*Populus canadensis*) (AGROFOREST-REHAB), and a remnant, old-growth forest (AGROFOREST-NATURAL), which was an undisturbed site with periodic flooding. All riparian buffers were adjacent to agricultural fields in corn-soybean rotations.

Soil characteristics were measured using three replicates per plot (n = 36), which were averaged to gain site level soil characteristics (S1). Soil samples were taken to a depth of 10 cm and kept at −20 °C until analysis. Soil at each location was classified as loam [21]. A portion of soil was dried at 105 °C to a constant mass, ground using a Retsch ball mill (Retsch, Düsseldorf, Germany), and analyzed for total soil N and C concentration (SN and SC, mg·g$^{-1}$) using approximately 0.1–0.15 g on a LECO CN628 (LECO Corporation, St Joseph, MI, USA). The other remaining portion of soil was

thawed and assessed for available inorganic N ($NO_3$ and $NH_4$, mg·kg$^{-1}$) using a Lachat QuikChem 8500 Series 2 Flow Injection Analyzer (Lachat Instruments, Loveland, CO, USA).

## 2.2. Plant Functional Traits

Within each buffer type, four $1 \times 1$ m plots were randomly established within 6 m of the waterline, along a 30 m parallel transect (n = 4 per buffer type per year). All herbaceous plants within the $1 \times 1$ m plots were identified using field manuals for species classification [22,23]. Based on the species identified in the plots, leaf samples were collected per species in mid-late July from plants just outside the plot to avoid interference with growth and soil disturbance. Each leaf was processed for morphological and chemical traits. Fresh leaf area was measured by scanning leaves and later analyzing images with ImageJ software [24]. Following scanning, leaves were dried at 65 °C to a constant mass. The specific leaf area (SLA) was calculated as a fresh leaf area/dry mass (m$^2$·g$^{-1}$). The leaves were ground into a homogeneous powder using a ball mill (Retsch Ltd., Haan, Germany). Leaf C (LCC, mg·g$^{-1}$) and N (LNC, mg·g$^{-1}$) concentrations were then measured on approximately 200 mg of dried leaf sample using a LECO CN628 (LECO Corporation, St Joseph, MI, USA). The ratio of carbon to nitrogen per leaf was used to calculate the leaf carbon/nitrogen ratio (L-C:N, mg·g$^{-1}$).

## 2.3. Plant Diversity Metrics

We calculated three types of plant diversity: taxonomic diversity, phylogenetic diversity, and functional diversity. Taxonomic diversity is reported as species richness (SR) and as Shannon's diversity index (TD). Phylogenetic diversity was determined by phylogenetic trees, which were created using Phylomatic [25] based on species abundance data from 2017 and 2018. These trees were then used to measure the evolutionary distance among species, employing the branch length adjuster (BLADJ) algorithm in Phylocom [26] and using the evolutionary ages provided by fossil records [27]. We report both Faith's phylogenetic diversity (PD) and Rao's quadratic entropy (QEphy) over both sampling years using the 'picante' package [28]. Functional diversity is reported as a single-trait functional diversity metric (FD) or as community weighted means (CWM). The abundance of species was determined by a quadrat sampling method, visually estimating the relative cover of each species. Single-trait FD metrics were calculated using aboveground traits (SLA, LNC, and L-C:N) in conjunction with the relative cover per species, per plot. We assessed total community trait variance, decomposed into, within, and between species trait diversity, following de Bello et al. (2011) [29]. Community weighted means (CWM) of leaf traits (SLA, LNC and L-C:N) were calculated using the average trait values for leaves (n = 12) per species within each riparian buffer type, where relative cover was used to represent the relative abundance of species within each plot.

## 2.4. Ecosystem Processes

Assessments of ecosystem processes, related to C and N cycling, were measured in the summer of 2017 and 2018 within each plot. Soil N-mineralization was quantified using polyethylene bags for in-situ incubations [30]. Soil samples were taken to a depth of 10 cm, homogenized, and placed into 2 bags, one of which was immediately frozen at −20 °C (T0) until analysis, whilst the other was placed back into the plot for an incubation time of 28 days (T28). Three replicates per plot were used per 28-day cycle (n = 36). To measure the available inorganic N ($NO_3$ and $NH_4$, mg·kg$^{-1}$), 2 g of soil was extracted with 20 mL potassium chloride (2M KCl), shaken for 30 min on an automated shaker table, and filtered using Q2 filter paper. All filtered samples were run through a Lachat QuikChem 8500 Series 2 Flow Injection Analyzer (Lachat Instruments, Loveland, CO, USA). The final result provided measurements of available inorganic nitrogen in forms of nitrate ($NO_3$-N) and ammonium ($NH_4$-N). N-mineralization was calculated as the difference between T0 and T28.

Greenhouse gas measurements were conducted bi-weekly to quantify the $CO_2$ (mg $CO_2$-C m$^{-2}$·h$^{-1}$), $N_2O$ (μg $N_2O$-N m$^{-2}$·h$^{-1}$), and $CH_4$ (μg $CH_4$-C m$^{-2}$·h$^{-1}$) efflux. PVC pipe collars, 10 cm in diameter and 25 cm in depth, were installed no more than 2 m away from each $1 \times 1$ m plot. The collars

were inserted to a depth of 10 cm, leaving a 15 cm headspace. Chamber caps were constructed from PVC covered with reflective insulation and contained a sampling port fitted with a rubber septum (1 cm diameter) for air extraction, as well a 10 cm long vent tube (9 mm inner diameter) to reduce pressure differences during sample collection (Dyer). Samples were taken at 0-, 15-, and 30-min intervals. Gas samples were removed from the chamber headspace and stored in 3 mL over-pressurized evacuated vials (LabCo Ltd., High Wycombe, UK) (Parkin and Venterea 2010). Bi-weekly gas sampling took place from mid-July to mid-August, a period of time corresponding to our leaf trait collection and species abundance measurements.

### 2.5. Statistical Analysis

Statistical analyses were performed using R statistical analysis software (R Foundation for Statistical Computing, Vienna, Austria). Prior to analysis, data were log transformed where necessary to obtain normality (CWM_SLA, CWM_L-C:N, FD_SLA, FD_L-C:N, TD, PD). We performed a one-way analysis of variance (ANOVA) to test for differences in herbaceous taxonomic (SR and TD), phylogenetic (PD and QEphy), and functional (CWM and FD) diversity among riparian buffer types. We used Tukey's post-hoc test to evaluate significant pair-wise differences.

We performed a principal component analysis (PCA) of FD indices and CMWs for three leaf traits (SLA, LNC and L-C:N) using the 'ade4' R package [31]. Based on these results, PCA axis 1 scores were used as a proxy for plant community diversity outcomes, representing functional diversity within single traits (FD) or the dominance of plant traits (CWM). Linear mixed-effects models were fit using the 'lme4' package to assess correlations using PCA axis 1 scores (FD or CWM) and soil gas flux and N-mineralization rates, with the year as a random effect. Soil $CO_2$, $N_2O$, and $CH_4$ efflux were log transformed prior to analysis to obtain normality. The 'MuMIn' package was used to obtain conditional $r^2$ data, describing the proportion of variance explained by our fixed factors as well as year as a random effect.

## 3. Results

### 3.1. Taxonomic and Phylogenetic Diversity of Plant Communities among Riparian Systems

We observed significant differences in taxonomic and phylogenetic diversity between the GRASSLAND and AGROFOREST-REHAB buffers and the AGROFOREST-NATURAL buffer ($p < 0.001$, Table 1). The highest species richness (SR) values were observed in the GRASSLAND (7.13 ± 0.52) and AGROFOREST-REHAB (8.88 ± 1.06) buffers, with the lowest species richness observed in the AGROFOREST-NATURAL buffer (3.63 ± 0.32). The Shannon diversity index (TD) show the same trend, with the highest diversity occurring in the GRASSLAND (1.53 ± 0.08) and AGROFOREST-REHAB (1.50 ± 0.07) buffers and the lowest in the AGROFOREST-NATURAL buffer (0.97 ± 0.09). Phylogenetic dissimilarity (QEphy) was also significantly different between the riparian types ($p < 0.001$, Table 1). Phylogenetic dissimilarity was highest in the AGROFOREST-NATURAL buffer (175.39 ± 28.45), followed by the AGROFOREST-REHAB (82.03 ± 3.91) and GRASSLAND buffers (78.35 ± 3.62).

**Table 1.** Mean (±SE) taxonomic (species richness and Shannon's diversity index) and phylogenetic (Faith's phylogenetic diversity and Rao quadratic entropy) diversity in three riparian buffer types (significant differences in bold).

| | | Riparian Buffer Type | | | F (*p*-Value) |
|---|---|---|---|---|---|
| | Diversity Indice | GRASS | AGROFOREST-REHAB | AGROFOREST-NATURAL | |
| Taxonomic diversity | Species richness | 7.13 ± 0.52 [a] | 8.88 ± 1.06 [a] | 3.63 ± 0.32 [b] | 14.36 (**<0.001**) |
| | Shannon's diversity index | 1.53 ± 0.08 [a] | 1.50 ± 0.07 [a] | 0.97 ± 0.09 [b] | 15.17 (**<0.001**) |
| Phylogenetic diversity | Faith's phylogenetic diversity | 537.84 ± 36.3 [a] | 763.81 ± 75.53 [a] | 735.14 ± 126.97 [a] | 1.18 (0.33) |
| | Rao quadratic entropy | 78.35 ± 3.62 [a] | 82.03 ± 3.91 [a] | 175.39 ± 28.45 [b] | 10.83 (**<0.001**) |

[a,b] Means in rows followed by the same letter are not significantly different at $p < 0.05$.

### 3.2. Functional Diversity of Plant Communities among Riparian Systems

Community weighted means (CWM) were significantly different among riparian buffer types ($p < 0.001$, Table 2). CWM SLA was lowest in the GRASSLAND buffers (0.02 ± 0.001), followed by the AGROFOREST-NATURAL (0.04 ± 0.001) and AGROFOREST-REHAB buffer (0.05 ± 0.001). As expected, the same trend was observed for CWM-LNC, with the lowest values measured in the GRASSLAND (25.23 ± 1.08) followed by the AGROFOREST-NATURAL (29.69 ± 0.68) and AGROFOREST-REHAB buffers (35.85 ± 0.55). The opposite trend was calculated for CWM-L-C:N, with the highest values measured in the GRASSLAND (17.93 ± 1.14) followed by the AGROFOREST-NATURAL (13.75 ± 0.29) and AGROFOREST-REHAB buffers (11.74 ± 0.27). Single trait functional diversity indices (FD) for SLA, L-C:N, and LNC were significantly different between riparian buffer types ($p < 0.001$, Table 2). Notably though, the lowest functional diversity for the SLA occurred in the GRASSLAND (0.005 ± 0.0003) followed by the AGROFOREST-NATURAL and AGROFOREST-REHAB buffers (0.01 ± 0.004 and 0.02 ± 0.002, respectively).

**Table 2.** Mean (±SE) plant functional trait diversity indices (community weighted means and functional diversity) within three riparian buffer types (significant differences in bold).

| Diversity Indice | | Riparian Buffer Type | | | F (*p*-Value) |
|---|---|---|---|---|---|
| | | GRASS | AGROFOREST-REHAB | AGROFOREST-NATURAL | |
| Community weighted means | SLA | 0.02 ± 0.001 [a] | 0.05 ± 0.001 [b] | 0.04 ± 0.001 [c] | 229.91 (<**0.001**) |
| | L-C:N | 17.93 ± 1.14 [a] | 11.74 ± 0.27 [b] | 13.75 ± 0.29 [c] | 24.74 (<**0.001**) |
| | LNC | 25.23 ± 1.08 [a] | 35.85 ± 0.55 [b] | 29.69 ± 0.68 [c] | 43.92 (<**0.001**) |
| Functional diversity | SLA | 0.005 ± 0.0003 [a] | 0.02±0.002 [b] | 0.01 ± 0.004 [a] | 16.37 (<**0.001**) |
| | L-C:N | 45.74 ± 11.31 [a] | 6.20 ± 0.56 [b] | 13.02 ± 2.23 [b] | 15.85 (<**0.001**) |
| | LNC | 55.72 ± 6.73 [a,b] | 74.47 ± 7.42 [a] | 38.93 ± 3.96 [b] | 8.18 (**0.002**) |

[a,b,c] Means in rows followed by the same letter are not significantly different at $p < 0.05$.

### 3.3. Multivariate Plant Community Variation

A principal component analysis (PCA) revealed that a singular axis (PCA axis 1) explained 53.9% of the variation in functional diversity (FD) in these riparian systems (Figure 1b). Clearly, GRASSLAND and AGROFOREST-REHAB are composed of distinct communities expressing dissimilar functional diversity. Similarly, a high proportion of the variation in CWM leaf traits is explained with a singular axis (92.5%) (Figure 1a) among plant communities in these riparian types. The leaf traits expressed within plant communities in the GRASSLAND and AGROFOREST buffers fall on opposite ends of a resource acquiring to resource conservative strategy axis. Communities with high LNC and SLA fall on one end of the spectrum, and communities with low LNC and SLA but high L-C:N fall on the other end of the spectrum. These values were significantly different between riparian buffer types ($p < 0.001$).

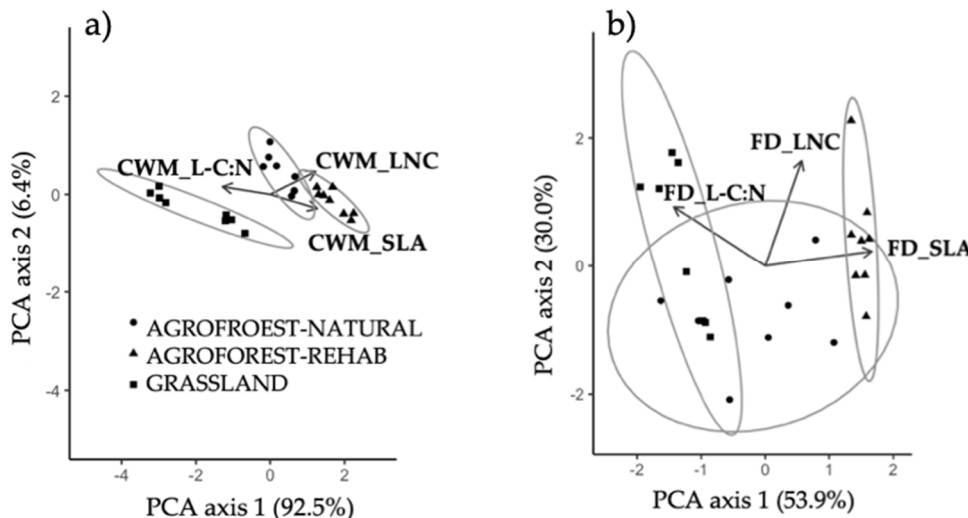

**Figure 1.** Principal component analysis using (**a**) three single trait (leaf N (LNC), leaf C:N (L-C:N), and leaf specific leaf area (SLA)) community weighted means (CWM) and (**b**) three single trait (LNC, L-C:N, SLA) functional diversity (FD) indices. The arrows represent the principal component loadings, and differences in riparian land use types are shown, where ellipses correspond to 95% confidence surrounding trait syndromes, within each riparian buffer type.

### 3.4. Relationships between Plant Community Diversity and Soil C and N Processes

We found significant differences in soil processes among our riparian buffer types (Table 3). Rates of soil N mineralization were significantly higher ($p \leq 0.01$) in the GRASSLAND and AGROFOREST-REHAB sites (8.5 ± 4.6 and 6.3 ± 2.7 mg·kg$^{-1}$ 28 d$^{-1}$, respectively) and were lowest in the AGROFOREST-NATURAL site ($-6.3 \pm 2.3$ mg·kg$^{-1}$ 28 d$^{-1}$). We also observed significantly higher rates of $CO_2$ efflux ($p < 0.01$) from soils in the GRASSLAND buffers (423.7 ± 67.7 mg $CO_2$-C m$^{-2}$·h$^{-1}$), followed by the AGROFOREST-REHAB and AGROFOREST-NATURAL buffers (190.5 ± 30.9 and 182.0 ± 42.5 mg $CO_2$-C m$^{-2}$·h$^{-1}$, respectively). Soil $CH_4$ efflux was also significantly higher in the AGROFOREST-NATURAL buffer (613.5 ± 164.7 µg $CH_4$-C m$^{-2}$·h$^{-1}$) compared to the AGROFOREST-REHAB and GRASSLAND buffers ($-0.9 \pm 26.4$ and $-92.9 \pm 45.7$ µg $CH_4$-C m$^{-2}$·h$^{-1}$, respectively). $N_2O$ efflux (µg $N_2O$-N m$^{-2}$·h$^{-1}$) was not significantly different among riparian buffer types.

**Table 3.** Mean (±SE) ecosystem processes (soil N mineralization (mg kg$^{-1}$ 28 d$^{-1}$), $CO_2$ (mg $CO_2$-C m$^{-2}$·h$^{-1}$), $N_2O$ (µg $N_2O$-N m$^{-2}$·h$^{-1}$), and $CH_4$ (µg $CH_4$-C m$^{-2}$·h$^{-1}$) efflux) across three riparian buffer types (significant differences in bold).

| Ecosystem Process | Riparian Buffer Type | | | F (*p*-Value) |
|---|---|---|---|---|
| | GRASS | AGROFOREST-REHAB | AGROFOREST-NATURAL | |
| N mineralization (mg kg$^{-1}$ 28 d$^{-1}$) | 8.5 ± 4.6 [a] | 6.3 ± 2.7 [a] | $-6.3 \pm 2.3$ [b] | 5.66 (**0.01**) |
| $CO_2$ efflux (mg m$^{-2}$ h$^{-1}$) | 423.7 ± 67.7 [a] | 190.5 ± 30.9 [b] | 182.0 ± 42.5 [b] | 6.04 (**<0.01**) |
| $N_2O$ efflux (µg m$^{-2}$ h$^{-1}$) | 19.0 ± 3.2 [a] | 32.9 ± 11.5 [a] | 24.8 ± 9.9 [a] | 0.72 (0.49) |
| $CH_4$ efflux (µg m$^{-2}$ h$^{-1}$) | $-92.9 \pm 45.7$ [a] | -0.9 ± 26.4 [a] | 613.5 ± 164.7 [b] | 12.65 (**<0.01**) |

[a,b] Means in rows followed by the same letter are not significantly different at $p < 0.05$.

Rates of soil $CO_2$ efflux were significantly and negatively related to plant community trait variation (Table 4: FD = $r^2$ = 0.19; $p$ = 0.05; CWM = $r^2$ = 0.26; $p$ = 0.01), with the highest rates occurring in the GRASSLAND buffers (Table 3). Rates of soil $CH_4$ efflux were significantly and positively related to FD ($r^2$ = 0.22; $p$ = 0.05; Table 4), with the highest rates occurring in the AGROFOREST-NATURAL buffer (Table 3). The rates of soil N mineralization and $N_2O$ efflux were not significantly correlated to multivariate plant community variation (Table 4).

**Table 4.** Linear mixed models, with the year as a random factor, describing the relationship between principal component analysis (PCA) scores derived from plant community variation (CWM and FD) and ecosystem functioning (mineralization, $CO_2$, $N_2O$, and $CH_4$ efflux). Slope estimates, $p$-values, and conditional $r^2$ are provided. Significant models are bolded.

| Multivariate Plant Community Variation | Ecosystem Process | Slope Estimate | $p$-Value | Cond. $r^2$ |
|---|---|---|---|---|
| Community weighted mean (CWM, 92.5% explained variation) | N Mineralization | −0.43 | 0.70 | 0.01 |
| | $CO_2$ efflux | **−0.22** | **0.01** | **0.26** |
| | $N_2O$ efflux | 0.01 | 0.94 | 0.30 |
| | $CH_4$ efflux | **0.35** | **0.05** | **0.22** |
| Functional diversity (FD, 53.9% explained variation) | N Mineralization | 0.42 | 0.78 | 0.004 |
| | $CO_2$ efflux | **−0.22** | **0.05** | **0.19** |
| | $N_2O$ efflux | −0.01 | 0.96 | 0.30 |
| | $CH_4$ efflux | 0.17 | 0.48 | 0.15 |

## 4. Discussion

### 4.1. Taxonomic and Functional Biodiversity in Riparian Agroforestry Systems

We observed significant differences in taxonomic diversity (SR and TD) and phylogenetic dissimilarity (QEphy) among riparian buffer types. These differences can largely be explained by successional patterns attributed to each individual buffer. In particular, the buffer derived from the old-growth forest (AGROFOREST-NATURAL) had the lowest taxonomic diversity compared with the grassland and the rehabilitated riparian buffer types. This pattern is expected as older forests tend to experience less disturbances, such as treefall gaps, which results in homogeneous environmental conditions that exclude herbaceous species [32]. While taxonomic diversity in agroforest buffers were low, phylogenetic dissimilarity among species was relatively high. This level of dissimilarity suggests high niche differentiation and niche filling within these riparian agroforestry buffers [33]. This has strong potential consequences for efficient nutrient capture during nutrient transport from agricultural fields to adjacent waterways.

While all three buffer types were composed of both native and non-native species, the most dominant species within the GRASSLAND buffer were invasive species (*Bromus inermis* and *Phalaris arundinacea* subsp. *Arundinacea*). Interestingly, a survey of a plant species at this study location was conducted in 1988 [34], and a comparison of the species reveals that these invasive species were not present 30 years ago [35], confirming that a rather rapid invasion in the grassland buffer took place. Incidents of invasion are often observed after land abandonment [36,37]. These invasive grass species were not observed within the agroforestry buffers. The diversity resistance hypothesis [38] suggests that more diverse plant communities reduce the ability of invasive species to establish themselves due to high competition [36]. Evidently, high herbaceous species diversity in the tree-based buffers contributes to lowering plant invasions. Broadly, our findings show that the compositional change of herbaceous species, over time, in agroforestry buffers will shift toward niche filling and potentially closed nutrient cycling, which are the key attributes of a highly functional riparian agroforestry buffer.

### 4.2. Riparian System Functional Diversity

Grassland riparian buffers were dominated by species expressing resource conserving leaf traits, such as low SLA and high leaf C:N community weighted means, both of which are traits that have been highly linked to decomposability [14,16]. In contrast, and as expected, the two tree-based agroforestry buffers were dominated by herbaceous communities expressing resource acquiring traits, such as a significantly higher SLA and leaf N levels. Previous research on both grassland and forest systems shows similar patterns in community structuring over time [39], with a clear divergence of communities falling along a spectrum of resource acquisition to resource conservation [16]. Soil resource availability strongly structures the expression of traits within communities of plants [40,41] where low nutrient availability often results in plants exhibiting resource-conserving traits. Low litter deposition in grasslands tends to minimize nutrient inputs [42], resulting in resource limited conditions. Indeed, the grassland buffer in our study was characterized by soils with significantly lower amounts of available N, which was in the form of $NH_4$, as compared to the agroforestry buffers (Table S1). The range of functional trait values within herbaceous communities among the buffer types showed similar trends. The more homogeneous grassland buffers were characterized by plants expressing low functional diversity in SLA and LNC, while high heterogeneity in the availability of light presumably contributed to a high diversity in leaf functional traits in the two agroforestry buffers.

### 4.3. Biodiversity and Ecosystem Processes in Riparian Agroforestry Systems

Decades of research has reported defined relationships between terrestrial system diversity and ecosystem functions [15,20], most notably within natural grasslands [43–46] and forests [47–50], using a range of diversity measures, including phylogenetic, taxonomic, and functional. Our findings offer some of the first on biodiversity–ecosystem function relationships in riparian agroforestry systems. We report correlations between FD and CWM indices and riparian ecosystem processes. Specifically, a higher herbaceous community diversity in functional traits and acquisitive trait dominance in the agroforestry buffers were significantly and negatively correlated to the rate of soil $CO_2$ efflux. Previous work has showed strong relationships between leaf traits and decomposition rates [51]. And one might expect that low SLA and high leaf C:N in the grassland may result in suppressed decomposition rates, however, we report the highest rates of soil $CO_2$ efflux from these systems in comparison to the other buffer types. This may be a product of two pathways: (i) higher decomposition in the grassland due to the high range of litter leaf C:N, contributing to litter quality conducive to decomposition, and thus higher measures of soil $CO_2$ flux, and/or (ii), though unmeasured, higher rates of root respiration in these fine-root dense grass dominant systems. Furthermore, other abiotic factors contribute to soil $CO_2$ efflux rates, such as the soil moisture dynamics in these plots near waterways and soil temperature fluxes under highly variable aboveground light dynamics [52].

While we did not find any significant correlations between $N_2O$ efflux and our functional diversity indices, previous studies, with experimentally manipulated grassland species, found correlations between acquisitive trait dominance and a reduction in $N_2O$ efflux, as a result of larger N uptake and biomass production [53]. In time, with changes in herbaceous community patterns within these buffer systems, significant effects on $N_2O$ efflux is expected. We also report a significant and positive correlation between functional diversity indices and rates of $CH_4$ efflux. This correlation was most likely dependent, again, on differences in soil moisture among the three riparian buffer systems (Table S1). The AGROFOREST-NATURAL site experienced prolonged flooding for the majority of the spring and summer, creating wetland-like conditions optimal for anaerobic processes (methanogenesis) [54]. And inversely, soil $CO_2$ and $N_2O$ efflux was suppressed because of the suboptimal conditions for the production of these gases. Trade-offs are apparent in the role of riparian system herbaceous communities in conditioning soils for various C- and N-based dynamics. Based on our findings, a functional traits approach provides indicators for estimating C and N dynamics, except under extreme soil conditions, where plant communities have less control than abiotic conditions on driving soil C and N dynamics in riparian systems.

## 5. Conclusions

Our findings contribute some of the first data on herbaceous community diversity patterns in riparian agroforestry systems. As management practices in riparian agroforestry systems focus on the manipulation of species composition, we show that shifts in community composition may indeed have a large impact on soil processes and the delivery of ecosystem services. Particularly with regards to soil C and N dynamics and greenhouse gas fluxes, herbaceous plant communities are strongly related to flux rates. Herbaceous communities with high functional diversity and resource acquiring trait syndromes, such as those in the agroforestry riparian systems, are significantly correlated with lower rates of soil $CO_2$ efflux. Therefore, the management of riparian agroforestry buffers not only contributes to the sequestration of carbon in trees and herbaceous biomass—these systems may also contribute to suppressed decomposition and therefore lower the $CO_2$ efflux from soils. These modified C pools and fluxes have important consequences for achieving carbon storage, such as targets laid out in the Intergovernmental Panel on Climate Change (IPCC) 2050 goals for carbon neutrality. Our findings provide further evidence that functionally diverse, and not necessarily taxonomically diverse plant communities are strongly correlated to positive ecosystem processes in agroforestry systems.

It is important to note that while our study identifies agroforestry as the best management practice for the reduction of $CO_2$ within riparian buffers, there are a host of other important functions, to which, diverse agroforestry riparian systems can contribute, including sediment and nutrient retention [55], enhanced biological pest control [56], pollination services [57,58], and the resistance to plant invasion [38]. In particular, the transformation of riparian systems into biologically complex, tree-based landscapes contributes to the creation of pockets of biodiversity within predominantly homogenous landscapes, and this addition of structural and functional complexity can provide resilience in the cases of local disturbances.

**Supplementary Materials:** The following are available online at http://www.mdpi.com/2071-1050/12/2/568/s1, Table S1: Site characterization of three riparian buffer types, with significant differences in bold.

**Author Contributions:** Conceptualization, S.W.B. and M.E.I.; methodology, S.W.B.; data collection: S.W.B. and M.B.; formal analysis, S.W.B.; resources, M.O., N.V.T., A.M.G., M.E.I.; writing—original draft preparation, S.W.B.; writing—review and editing M.B., M.O., N.V.T., A.M.G., M.E.I.; supervision M.E.I.; funding acquisition, N.V.T., M.E.I. All authors have read and agreed to the published version of the manuscript.

**Funding:** Funding for this project was from Agriculture and Agri-Food Canada's Agriculture Greenhouse Gases Program, and a Natural Sciences and Engineering Research Council of Canada (NSERC) Discovery Grant to M.E.I.

**Acknowledgments:** We would like to thank the landowners who so graciously allowed us to research on their land, as well as Kira Borden and Ato Aggrey for field assistance during the summer months.

**Conflicts of Interest:** The authors declare no conflict of interest.

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
