# Peer review of "Plant Diversity and Agroecosystem Function in Riparian Agroforests: Providing Ecosystem Services and Land-Use Transition"

_sustainability, doi:10.3390/su12020568_

Round 1

Reviewer 1 Report

Comments to the paper manuscript, entitled „Integrating plant functional diversity and agroecosystem functioning within riparian agroforestry systems “

The manuscript provides valuable data on functional diversity within riparian agroforestry systems.

There are not any serious shortcoming in this paper.

I should like to comment briefly on the following points.

Keywords. The list of keywords is missing identification of investigated ecosystems, e.g.  riparian ecosystems or agroforestry.  

Material and methods. This section is adequately described. However authors should specify

which crops were adjacent to the each riparian buffer type, soil types at each riparian buffer type (World reference base for soil resources), grassland management after conversion to natural grassland.

The authors could also mention that the site AGROFOREST–NATURAL has been flooded.

Results and discussion is well written. However the presence of invasive species in GRASSLAND buffer type can be a result of “abandonment” of management.

Conclusions are supported by the results.

Kind regards

Author Response

Thank you for your insightful comments and the opportunity to revise the manuscript.  Our responses to each comment are below.

Reviewer 1

point 1: Keywords. The list of keywords is missing identification of investigated ecosystems, e.g.  riparian ecosystems or agroforestry.  

Response 1: Thank you for this recommendation. “Riparian agroforestry” has been added to our key word list. 

point 2: authors should specify which crops were adjacent to the each riparian buffer type, soil types at each riparian buffer type (World reference base for soil resources), grassland management after conversion to natural grassland.

Response 2: Thank you for these suggestions. We have now added the type of crop adjacent to the riparian buffer systems (line 87). We have augmented our analysis with soil textural classification which is now noted on line 90. Finally, we point the reviewer to line 78-79 for a description of previous grassland management (“pastureland which had previously (over 30 years ago) been fenced off and returned to a natural grassland”).

point 3: The authors could also mention that the site AGROFOREST–NATURAL has been flooded.

Response 3: We have added this to our description of the agroforestry-natural site (“an undisturbed site with periodic flooding” line 86).

point 4: Results and discussion is well written. However the presence of invasive species in GRASSLAND buffer type can be a result of “abandonment” of management.

Response 4: Thank you for pointing this out. We have included a statement on the effects of abandonment on invasive species within the discussion with new appropriate citations to the invasion ecology literature.

Reviewer 2 Report

Add a section on how network agroecology can be used to design systems that maximize ecosystem services

In conclusion a section on the 2050 target on carbon neutral impact on he interface of agricultural landscapes and aquatic systems management.

Author Response

Thank you for your insightful comments and the opportunity to revise the manuscript.  Our responses to each comment are below.

Reviewer 2

point 1: Add a section on how network agroecology can be used to design systems that maximize ecosystem services

Response 1:  Thank you for this suggestion and we concur that this paper offers an opportunity to discuss agroecological design. We have now added new context throughout the manuscript in order to place this work in a larger, networked agroecology and landscape transformation context. Specifically, we draw the reviewers attentions to lines 14-17, 34-35, 39-41, 328.

point 2: In conclusion a section on the 2050 target on carbon neutral impact on the interface of agricultural landscapes and aquatic systems management.

Response 2: Thank you for the important suggestion. A new discussion has been added to the conclusion section on the role of agroforestry riparian buffers in achieving 2050 carbon neutrality goals. Line 317-321.